# Effects of Short-Term Inhibition of Rho Kinase on Dromedary Camel Oocyte In Vitro Maturation

**DOI:** 10.3390/ani10050750

**Published:** 2020-04-25

**Authors:** Hammed A. Tukur, Riyadh S. Aljumaah, Ayman Abdel-Aziz Swelum, Abdullah N. Alowaimer, Mutassim Abdelrahman, Islam M. Saadeldin

**Affiliations:** 1Department of Animal Production, College of Food and Agricultural Sciences, King Saud University, Riyadh 11451, Saudi Arabia; tukurhammeda@gmail.com (H.A.T.); rjumaah@ksu.edu.sa (R.S.A.); aswelum@ksu.edu.sa (A.A.-A.S.); aowaimer@ksu.edu.sa (A.N.A.); amutassim@ksu.edu.sa (M.A.); 2Department of Theriogenology, Faculty of Veterinary Medicine, Zagazig University, Zagazig 44519, Egypt; 3Department of Physiology, Faculty of Veterinary Medicine, Zagazig University, Zagazig 44519, Egypt

**Keywords:** Rho kinase, Y-27632, oocytes, meiosis, in vitro maturation, camel

## Abstract

**Simple Summary:**

Our results revealed, for the first time, that short-term inhibition of Rho-associated protein kinases (ROCK) for 4 h prior to in vitro maturation (IVM) in a biphasic IVM approach improved oocyte nuclear maturation, producing more MII oocyte, through modulating the expression of cytokinesis- and antiapoptosis-related mRNA transcripts. This positive result suggests ROCK inhibitor as a potential candidate molecule to exploit in the control of oocyte meiotic maturation.

**Abstract:**

This is the first report on a biphasic in vitro maturation (IVM) approach with a meiotic inhibitor to improve dromedary camel IVM. Spontaneous meiotic resumption poses a major setback for in vitro matured oocytes. The overall objective of this study was to improve in vitro maturation of dromedary camel oocytes using ROCK inhibitor (Y-27632) in a biphasic IVM to prevent spontaneous meiotic resumption. In the first experiment, we cultured immature cumulus–oocyte complexes (COCs, *n* = 375) in a prematuration medium supplemented with ROCK inhibitor (RI) for 2 h, 4 h, 6 h, and 24 h before submission to normal in vitro maturation to complete 28 h. The control was cultured for 28 h in the absence of RI. In the first phase of experiment two, we cultured COCs (*n* = 480) in the presence or absence (control) of RI for 2 h, 4 h, 6 h, and 24 h, and conducted real-time relative quantitative PCR (qPCR) on selected mRNA transcripts. The same was done in the second phase, but qPCR was done after completion of normal IVM. Assessment of nuclear maturation showed that pre-IVM for 4 h yielded an increase in MII oocyte (54.67% vs. 26.6% of control; *p* < 0.05). As expected, the same group showed the highest degree (2) of cumulus expansion. In experiment 2, qPCR results showed significantly higher expression of *ACTB* and *BCL2* in the RI group treated for 4 h when compared with the other groups. However, their relative quantification after biphasic IVM did not reveal any significant difference, except for the positive response of *BCL2* and *BAX/BCL2* ratio after 4 and 6 h biphasic IVM. In conclusion, RI prevents premature oocyte maturation and gave a significantly positive outcome during the 4 h treatment. This finding is a paradigm for future investigation on dromedary camel biphasic IVM and for improving the outcome of IVM in this species.

## 1. Introduction

Reproductive success in livestock ensures the continuation, survival, and conservation of species, and, by extension, food security. To maximize productivity and to address several reproductive challenges, assisted reproductive technologies (ARTs) have come to the rescue [1,2,3]. The dromedary camel (*Camelus dromedarius*), with its natural ability to produce quality meat, milk, and fiber under very hot and hostile climatic conditions [4], is reproductively weak [3], and has received the least attention among livestock [5,6]. Few important updates on ARTs in camel were recently reviewed [3,6], and as stated, few research centers are working on camel reproduction, and mostly on male camels. However, camel oocyte research is fallow and the success rate of ARTs is very low [7].

Studies on camel oocytes are still unfolding. So far, several attempts to improve camel oocyte maturation have been made [8,9,10]. One major challenge that hinders camel oocyte maturation is the spontaneous meiotic resumption that occurs when immature oocytes transfer from the natural follicular environment to be cultured in vitro. This drastic change causes oocytes to spontaneously resume meiosis due to the absence of inhibitive signals [11,12]. This sudden meiotic resumption causes loss of cumulus–oocyte communication [13], which is crucial to oocyte health, metabolism, and acquisition of developmental competence [14]. Eventually, oocytes will have asynchronous premature nuclear and cytoplasmic maturation, which leads to premature oocyte development. Even though these oocytes appear morphologically normal, are not competent. This issue has been the subject of numerous earlier and recent studies [15,16,17].

The biphasic IVM approach involves a temporary delay in meiosis resumption (holding the oocyte at the GV stage in vitro) in a prematuration phase using meiotic inhibitors before submission to normal in vitro maturation [16,18]. It is hypothesized that this temporary delay of meiotic resumption could mimic the natural follicular environment and improve the developmental competence of in vitro matured oocytes [17].

Several approaches have been examined to delay oocyte meiotic resumption. These include modulation of cAMP signaling (e.g., C-type natriuretic peptide, forskolin, and isobutylmethylxanthine) [16,19,20,21], inhibition of key molecules (e.g., butyrolactone I and roscovitine) [22,23,24,25], and use of adenosine [17].

Rho-associated protein kinases (ROCK) maintain embryonic stem cells properties, promoting their recovery and their post-thaw proliferation [26,27]. Recently, ROCK inhibitor (RI, Y-27632) has been implemented in in vitro production of human embryos [28] to increase the revivability of blastocysts, and even for oocytes, after vitrification [29,30]. Studies have described the use of Y-27632, a selective inhibitor of Rho-associated protein kinases (ROCK), as an active agent inhibiting oocyte meiotic resumption [24,31]. The Rho kinase is involved in many cellular functions and has been shown to be important for oocyte meiotic progression and embryonic development [29].

The role of ROCK in dromedary camel oocyte maturation and the potential effect of its inhibition to prevent spontaneous meiotic maturation in camel IVM have not been previously investigated. The aim of the present study, therefore, is to improve the in vitro maturation of dromedary camel cumulus–oocyte complexes using ROCK inhibitor (Y-27632) in biphasic IVM to prevent spontaneous meiotic resumption.

## 2. Materials and Methods

### 2.1. Media and Reagent

Unless otherwise stated, reagents were obtained from Sigma-Aldrich Corp. (St. Louis, MO, USA). Media preparation and the experimental procedure were performed as described by [9,32,33] unless otherwise stated.

### 2.2. Ovary Collection and Oocyte Retrieval

Camel ovaries (*n* = 320) were collected from a local slaughterhouse in Riyadh and transported within 3 h after slaughter to the laboratory in thermos flasks containing prewarmed saline 0.9% (*w*/*v*) maintained at 33–36 °C. Fluid from antral follicles (2–8 mm) was aspirated using an 18-gauge needle connected to a 10 mL syringe to obtain the cumulus–oocyte complexes (COCs). The aspirate was placed in a 60 mm sterile IVF dish. Then, COCs wash medium composed of tissue culture medium 199 (TCM-199) supplemented with 2 mM NaHCO_3_, 0.1% bovine serum albumin (BSA), 10% (*v*/*v*) fetal bovine serum (FBS), 5 μg/mL gentamycin sulfate, and penicillin/streptomycin was added and left to settle for 30 s. COCs with intact, uniform cytoplasm, and compact cumulus were selected under a stereomicroscope and randomly distributed for the experiments.

### 2.3. Experiment Design and Procedure

#### 2.3.1. Experiment 1: Biphasic IVM with ROCK Inhibitor

Phase 1: Pre-IVM incubation period with RI for different durations (Figure 1).

Selected COCs were transferred into new 60 mm dishes containing wash medium (mentioned above) for further washing. Washing was done three times. The COCs were further examined, selected and transferred to the pre-IVM culture medium that was supplemented with 10 µM Y-27632 (RI) to block meiotic resumption [28,34]. Together with RI, the pre-IVM medium was composed of (TCM-199) supplemented with 10% (*v*/*v*) sterile camel follicular fluid, 10% fetal bovine serum (FBS), 10 μg/mL follicle stimulating hormone (FSH), 10 μg/mL luteinizing hormone (LH), 1 μg/mL 17β-estradiol, 20 ng/mL epidermal growth factor (EGF), 1 μL/mL insulin-transferrin-selenium (ITS), 0.3 μM cysteamine, 0.15 mg/mL L-glutamine, 1 μL/mL gentamycin sulfate and Sodium pyruvate. Then, COCs were incubated in a humidified chamber at 38 °C for different time periods (2 h, 4 h, 6 h, and 24 h). The control group was incubated without RI. Each group contained 15 COCs that were randomly selected. The experiment was replicated five times. The total number of COCs used was 375 in a completely randomized design.

Phase 2: Normal IVM

After pre-IVM for the respective treatments, all COCs were transferred into normal IVM medium (without RI supplement) in separate wells to complete the 28 h maturation period—the optimum time for dromedary camel oocyte maturation in vitro [9]. Treatment group RI 2 h was transferred into normal maturation medium and was matured for 26 h. Treatments RI 4 h, 6 h, and 24 h were transferred and matured for 24 h, 22 h, and 4 h, respectively (Figure 1). Thereafter, COCs of each group were collected and examined for meiosis I completion (first polar body extrusion), cumulus expansion, and oocyte morphometric evaluation.

##### Evaluation of Cumulus Cell Expansion after Biphasic IVM

The degree of cumulus expansion was assessed at the end of IVM culture as described previously [8]. Briefly, cumulus layers were observed under a microscope and graded as 0, 1, or 2. Grade 0 indicated no expansion, with cumulus cells compacted around the oocyte. Grade 1 represented partial cumulus expansion. Grade 2 represented complete expansion of the cumulus layer, including the corona radiata.

##### Assessment of Oocytes for Completion of Meiosis I

To assess the oocytes for the completion of meiosis I (extrusion of first polar body), COCs were denuded (stripped of cumulus cells) chemically by repeated pipetting in TCM-199 supplemented with 1 mg/mL hyaluronidase. This was done separately for each group. Denuded oocytes were washed three times in wash medium (TCM-199 supplemented with 10% FBS and penicillin/streptomycin). Thereafter, cumulus free oocytes were stained with 5 μg/mL bisbenzimide (Hoechst 33342) for 5 min to visualize the nuclear materials and polar body extrusions using an inverted microscope equipped with epifluorescence (Leica DMI4000 B, Leica Microsystems GMS GmbH, Wetzlar, Germany). Oocytes were classified as matured (extrusion of first polar body, Metaphase II), immature (condensed nuclear materials with no polar body), and degenerated (no nuclear materials).

##### Oocyte Morphometric Evaluation

Oocyte morphometrics, including ooplasm diameter, zona pellucida thickness, oocyte diameter, and perivitelline space, were measured in three different areas of the oocyte using ImageJ 1.50i software (NIH, Bethesda, MD, USA) (Appendix A). Finally, the mean values were calculated and were used in further analysis. Measurements were as described by Saadeldin et al. [32].

#### 2.3.2. Experiment 2

Effect of biphasic IVM with RI on expression of cytokinesis- and apoptosis-related mRNA transcripts of dromedary camel COCs.

##### Collection of COCs for RNA Isolation

To determine the effect of RI on the expression of mRNA transcript of selected apoptotic- (*BCL2*, *BAX*, and *CTSB*) and cytokinesis-related genes (*ACTB*, *TUBA1A*, and *CFL1*), COCs for each group were collected in two phases:

Phase 1: Here, COCs were collected after pre-IVM. Briefly, we retrieved 480 fresh COCs from 500 ovaries as described in experiment 1. Then, COCs were randomly distributed into the RI-supplemented group or a control group. There were four independent treatment groups, each with an individual control. The treatment groups were incubated in pre-IVM medium supplemented with RI as described above, while the control groups were in normal IVM medium for the same duration as their corresponding treatments. After pre-IVM, COCs of both control and the corresponding treatment groups were collected. COCs of the RI 2 h group and its control were collected after 2 h of incubation. The COCs were collected in a solution of phosphate buffered saline (PBS) in 1.5 μL sterile tubes and snap frozen at −80 °C until RNA extraction was done. The same was done for other groups and their respective controls.

Phase 2: COCs were collected after biphasic IVM and randomly distributed into four groups (one control and three treatments). After pre-IVM for 2 h, 4 h, and 6 h, COCs were transferred into normal IVM medium to complete 28 h incubation. The control groups were matured in normal IVM medium without RI. For all groups, COCs were collected after 28 h of incubation. A total of 240 COCs was used, with 20 COCs per group. The experiment was replicated at least three times.

##### Total RNA Isolation, Complimentary DNA Synthesis, and Real-Time PCR

Total RNA was isolated from the COCs collected for each group using the innuPREP RNA Mini kit (Analytik Jena AG, Jena, Germany) following the manufacturer’s guide. The concentration and quality of the total RNA was estimated using a NanoDrop™ 2000 spectrophotometer (Thermo Fisher, Waltham, MA, USA). High-capacity cDNA Reverse Transcription Kit (Applied Biosystems, Foster City, CA, USA) was used to synthesize cDNA (20 µL reaction volume) from the total RNA samples according to the manufacturer’s instructions. Briefly, RT master mix sufficient for 10 samples was prepared on ice. The components included 10 μL (100 ng) RNA, 2.0 μL of 10 × RT buffer, 0.8 µL of 25× dNTP mix (100 mM), 2.0 µL of 10× RT random primers, 4.2 µL of nuclease-free water, and 1.0 µL MultiScribe™ Reverse Transcriptase. RT reactions were run in four steps at 25 °C for 10 min, 37 °C for 120 min, then 85 °C for 5 min, and finally at 4 °C until the plate was removed. The cDNA samples were then stored at −20 °C until further reaction. Relative quantification of mRNA transcripts was performed using real-time PCR (Applied Biosystems). The reactions contained 1 μL (100 ng) cDNA, 1 µM forward primer, 1 µM reverse primer, 7 μL nuclease-free water, and 10 μL SYBR^®^ Green Premix (Applied Biosystems). Three replicates for each sample and primer were run. The fold change and relative quantification of cytokinesis or morphology-related (*ACTB*, *TUBA1*, and *CFL1*) and apoptotic-related (*BCL2*, *BAX*, *CTSB*) transcripts were normalized to the house-keeping gene *GAPDH* and calculated with the 2^−ΔΔCt^ method described by [35] in relation to the reference gene and the reference group. Thermocycling conditions were 95 °C for 10 min initial denaturation, followed by amplification at 40 cycles of 95 °C for 10 s, 60 °C for 20 s, and 72 °C for 40 s. Primers were designed using the Primer-3 online tool and camel-specific (*Camelus dromedaries)* GenBank sequences. Details of the primer set used are presented in Table 1.

#### 2.3.3. Statistical Analysis

COCs were randomly assigned to the experimental groups, and the experiments were repeated at least five times. An unpaired *t*-test was used when the means of two groups were compared. When more than two conditions were being compared, data were analyzed using one-way ANOVA. When a significant difference was identified (*p* < 0.05), individual treatment differences were compared using Tukey-HSD test. All statistical analyses were carried out using R software (version 3.6.2) (R Core Team) with R Studio editor using the *Agricolae* package. Microsoft Excel was used to generate graphs. Pearson’s linear correlation coefficients were calculated to determine the correlation (r) between the means of oocyte maturation indices, morphometric parameters, and mRNA transcript expression, where r values > ±0.7 were considered strong positive/negative linear relationships, r > ±0.5 were considered moderate positive/negative linear relationships, and r <  ±0.5 were considered weak positive/negative linear relationships [36].

## 3. Results

### 3.1. Effect of RI on Camel IVM, Oocyte Morphometry, and Cumulus Expansion

RI showed no impact on the cumulus morphology when COCs were cultured for 2, 4, 6 h when compared with control group (Appendix A). For cumulus expansion (Table 2, Figure 2), the 4P group showed fully expanded cumulus cells (grade 2), while the rest of the treatments and the control all showed partial (grade 1) expansion.

The first experiment determined the effect of biphasic IVM (pre-IVM with RI for different duration followed by normal IVM for the remaining maturation time) on dromedary camel COC maturation. The COC maturation parameters examined were first polar body extrusion (meiosis I completion), cumulus morphology and expansion, and oocyte morphometry. The results shown in Table 2 indicate that pre-IVM duration in a biphasic IVM with RI has significant effect on COC maturation. For the 4 h pre-IVM with RI (4P), the highest incidence of first polar body extrusion was obtained (54.67%). The percentage of degenerated oocytes (17.3%) and no polar body (immature) oocytes (28%) were the lowest. Prolonged inhibition of COCs with RI for 24 h (24P) maintained meiotic arrest and hence these cells had the lowest incidence of first polar body (4%), and the highest incidence of degenerated oocytes and oocytes with no first polar body. Patterns of nuclear maturation are shown in Figure 3.

To explore the effect of biphasic IVM with ROCK (pre-IVM with ROCK inhibitor for varying duration) on oocyte morphometry, COCs were denuded after IVM and observation and image capturing were achieved using an inverted microscope. The captured images were used to obtain morphometric parameters including oocyte diameter (OCD), zona pellucida thickness (ZP), length of perivitelline space (PS), and ooplasm diameter (OPD) (see Appendix A). As presented in Table 3, significant difference (*p* < 0.05) was observed in all groups for all the parameters measured. The 6P group showed the smallest oocyte diameter (138.82 ± 1.11 µm), zona pellucida thickness (9.14 ± 0.25 µm), and ooplasm diameter (103.93 ± 0.7 µm). The control group had the thickest zona pellucida (12.97 ± 0.5 µm). The perivitelline space (PS) length was smaller in the control group (2.56 ± 0.41 µm) and the 4P group (3.19 ± 0.30 µm), while large PS length was observed in both the 2P (6.01 ± 0.30 µm) and 6P (6.55 ± 0.7 µm) groups. The ooplasm diameter in the 2P, 4P, and control group were not significantly different.

### 3.2. Effect of Biphasic IVM with RI on the Expression of Cytokinesis- and Apoptosis-Related mRNA Transcripts

First, we examined the immediate effect of RI on the expression of mRNA transcript of genes mentioned in Table 1. COCs were collected for RNA extraction immediately after the prematuration phase (normal IVM was not continued here). In the 2 h RI (2P, Figure 4A), we observed *ACTB* to be the only cytokinesis gene with markedly reduced expression level. A simultaneous reduction of both of proapoptotic transcripts (*BAX*, *BAX/BCL2* ratio) and antiapoptotic *BCL2* was surprising. After inhibiting ROCK for 4 h (4P), we found a positive effect on the expression of cytokinesis related genes (except for *TUBA1* which was significantly reduced) and anti-apoptotic *BCL* in 4P (Figure 4B). The ratio of *BAX/BCL2* was significantly reduced compare to the 4C control group (Figure 4B). The expression level of *BCL2* was reduced in the 6 h (6P) group (Figure 4C). When we inhibited ROCK for 24 h (24P), we observed a clear negative response. Antiapoptotic (*BCL2*) transcripts were significantly lowered, while proapoptotic (*BAX*) transcripts had significant increase (Figure 4D).

Next, we examined the effect of biphasic IVM (pre-IVM with RI for 2 h (2P), 4 h (4P), and 6 h (6P) followed by normal IVM (as illustrated in Figure 1)) on the expression of the same cytokinesis-related (*ACTB*, *TUBA1*, and *CFL1*) and apoptosis-related (*BCL2*, *BAX*, and *CTSB*) genes. There was no significant difference (*p* > 0.05) in the relative quantification of *ACTB, CFL1, BCL2*, and *CTSB* amongst treatments (Figure 5). Biphasic IVM with RI for 2 h pre-IVM (2P) significantly showed increased *TUBA1* and *BAX* (*p* < 0.05) compared to the other groups.

### 3.3. Correlation Analysis between Oocyte Maturation, Oocyte Morphometry, and Different mRNA Transcripts

As shown in Table 4, oocyte maturation is directly correlated with cumulus expansion; however, they are both negatively correlated with oocyte degeneration. Moreover, oocyte diameter is directly correlated with ooplasm diameter (r = 0.94). We further associated mRNA transcript expression with oocyte maturation or polar body extrusion and oocyte morphometry and found *ACTB* to be positively correlated with oocyte maturation (r = 0.65), cumulus expansion (r = 0.77), oocyte diameter (r = 0.68), and ooplasm diameter (r = 0.65). Similarly, for *CFL1*, there was positive correlation with oocyte maturation (r = 0.84), cumulus expansion (r = 0.52). *TUBA1* showed inverse correlation with cumulus expansion (r = −0.65) and strong positive correlation with oocyte degeneration (r = 0.77). *BAX* and *BAX/BCL2* ratio both showed negative correlation with polar body extrusion (r = −0.56 and −0.59, respectively) and cumulus expansion (r = −0.58), with a positive correlation with oocyte degeneration (r = 0.7 and 0.71, respectively). The relationship between *BCL2* and oocyte diameter and ooplasm diameter was positive (r = 0.98 and 0.87, respectively). For interrelations of mRNA transcript expression, ACTB directly correlated with BCL2 (r = 0.72) which showed direct correlation with TUB (r = 0.61). TUB showed direct correlation with BAX (r = 0.96) and inverse correlation with CFLN (r = 0.55). On the other hand, CTSB showed no relation with the studied genes, while it showed positive correlation with cumulus expansion (r = 0.58) and ooplasm diameter (r = 0.65), with negative correlation with oocyte degeneration (r = 0.57), which indicates a tendency for COCs to undergo apoptosis by the end of IVM stages.

## 4. Discussion

Since the recognition of in vitro maturation as a simple, promising, noninvasive, and cheaper procedure for embryo production in ARTs, extensive studies have been conducted on different species. Among livestock, cattle and pigs have both received the most attention in this area [37]. In the recent years, notable improvement and advances in IVM techniques have been demonstrated. The different biphasic IVM methods reported in recent studies have recorded amazing outcomes in humans [18,38] and livestock species [17,39]. All these studies used varying doses of meiotic inhibitors for different pre-IVM durations. The procedure used in the present study examined the effect of pre-IVM duration using the same dose of meiotic inhibitor, RI.

Few studies have investigated the role of ROCK signaling in the maturation medium of oocytes in some animal species. Our findings identified ROCK as a key player in dromedary camel oocyte meiotic maturation as in the pigs [24,40], cattle [41], cats [29], caprinae [30,42], and mice [31,43]. Oocyte maturation is a crucial process through which oocytes acquire their intrinsic capacity to support fertilization and early embryo development. This is also, by extension, crucial for the reproduction and survival of species.

Our studies showed the importance of ROCK in cytokinesis and regulation of dromedary camel oocyte nuclear maturation. We observed that long-term inhibition by ROCK (6 and 24 h RI biphasic IVM) resulted in lower incidence of polar body development and incomplete meiosis I. This indicates a gradual failure in the migration of the spindle to the cortex as a result of prolonged inhibition. As reported earlier, activation of ROCK causes increased cell migration, while the inhibition of ROCK signaling inhibits cell migration [44]. Our results clearly suggest that 24 h inhibition of ROCK may have detrimental effects on camel oocyte meiotic progression.

It is well known that optimal cumulus cell expansion is essential for normal oocyte maturation, oocyte cytoplasmic maturation in particular [45,46]. Impaired cytoplasmic maturation was associated with poor cumulus cell development [45,47]. The data in this study showed that cumulus cell expansion reached a maximum, with visible corona radiata, after 4RI biphasic IVM. This is a clear indication that oocyte maturation occurred more in the 4RI group. Cumulus expansion is positively linked to oocyte maturation [48]; therefore, the partial expansion and incomplete meiosis I in the other groups could mean that oocyte maturation was somehow interrupted, perhaps by the prolonged delay in meiotic resumption (in the case of 6 and 24 RI biphasic IVM group) or early resumption (in the case of the control and 2 RI biphasic IVM group). As expected, prolonged arrest and spontaneous or early meiotic resumption negatively affect the overall development of the oocyte. Taken together, these observations could partly explain the highest number of fully matured oocyte (MII oocytes) being observed in the 4RI biphasic IVM group.

Our results revealed significant differences in oocyte morphometry for the various treatment groups, and these values were comparable to our previous findings [32]. In the report, we observed that oocytes with no nuclear material had reduced ooplasm diameter. Our present results support this finding and indicate a possible correlation between reduced ooplasm and polar body extrusion failure. This also corroborates the reports of Ferrarini Zanetti et al. [49], in which increased perivitelline space and loss of gap junctions with the cumulus cells were linked with reduced oocyte competence. The exceptional positive effect of *ACTB* on polar body extrusion, oocyte and ooplasm diameter, and cumulus expansion showed the central role of β-actin in oocyte meiosis, spindle rotation, and cellular morphology. Moreover, *TUBA1* increased with oocyte diameter, and CFLN increased with oocyte maturation and cumulus expansion, which supports the necessity of cytokinesis-related transcripts in the final stages of COC maturation. The resulting alterations and abnormalities in oocyte diameter and PB extrusion by RI are due to the direct effect of ROCK on tubulin and other cytoskeleton proteins’ organization [31,42,44,50,51,52,53,54,55,56]. Moreover, ZP thickness decreased with increasing RI supplementation duration. It is known that ZP thickness reduces with progression of oocytes from the germinal vesicle stage to MII [57]; therefore, the observed increase in thickness in the control group might be related to the inhibitory effects of RI, but further studies are required to confirm this.

While the highest rate of nuclear maturation was observed after biphasic IVM with 4 h RI, there was no distinction in the level of mRNA transcripts of cytokinesis genes studied across all the groups (2 h, 4 h, 6 h RI biphasic IVM and Control). Considering the result of nuclear maturation, it would be expected that *ACTB*, *TUBA1*, and *CFL1* would show the highest transcript levels in the 4 h RI biphasic IVM group. Interestingly, the expression of *ACTB* and *BCL2* were distinctly higher when we examined them immediately after the 4 h pre-IVM period (Figure 4). This unparalleled observation could suggest that the biological function (mechanism) of ROCK, especially its effect on these genes, slightly differs in the dromedary camel oocyte. Future research should consider this for better understanding.

Reports on ROCK for porcine [40], bovine [41], and mouse [31] IVM identified a profound effect of RI on actin-mediated spindle organization and migration for polar body formation. Actin filaments are the major backbone of oocyte asymmetric spindle positioning, which results in polar body extrusion during meiosis [58,59,60]. However, it is important to note that spindle organization and polar body extrusion are not coordinated by a single gene. Many cytokinesis-related proteins, including actin, formin, profilin, cofilin, myosin, and tubulin, are effectors of the ROCK signaling pathway though which spindle positioning is regulated [43,52,53,54,61,62,63,64]. Moreover, ROCK inhibitor increased cellular microtubule acetylation because ROCK regulates microtubule acetylation via phosphorylation of tubulin polymerization-promoting protein 1 [44,50]. Meanwhile, our study targeted *ACTB* (beta-actin) from the actin family, whereas there are other members of the actin family (such as F- and G-actin) that could exert similar or greater effects on oocyte meiosis if ROCK is inhibited. Some reports considered the contribution of F-actin to be greater [58]. In the pig oocyte, G-actin was found in abundance in the cytoplasm at all stages, while F-actin was distributed in the cortex of oocytes [61].

ROCK signaling is a regulatory network for prevention of mitochondrial injury and apoptosis [65,66]. Some studies have highlighted antiapoptotic effects of RI on oocytes that could rescue the damaging effects of vitrification [30,67]. The ratio of *BAX/BCL2* reflects the balance between apoptotic and antiapoptotic signals and determines the cell’s susceptibility to apoptosis and the fate of cell survival [68,69,70]. The observed effect of RI on *BAX/BCL2* could also possibly explain the higher rate of nuclear maturation and fully expanded cumulus in the 4 h RI group. A higher level of *BCL2* is able to repress apoptosis, while overexpression of *BAX* accelerates cell death by apoptosis. The high level of *BAX* (Figure 5) may have overshadowed the contribution of other important genes that regulates cytokinesis and oocyte survival during maturation. It is therefore not surprising to have observed extremely low incidence of polar body extrusion in the other groups. Both degenerated and no-polar-body oocytes point to failure of the oocyte to exit the germinal vesicle stage and resume meiosis I.

The action and interaction of various related proteins or their combined effect could be species- dependent. For example, in goat oocytes, RI decreased the maturation rate from 45% (Control) to 42%, but when there is special combination of RI with cysteamine and leukemia inhibitory factor, the maturation rate was improved to 67% [30], owing to the synergistic effects of RI and the prevention of apoptosis. 

## 5. Conclusions

Our results demonstrated, for the first time, that ROCK is involved in dromedary camel oocyte maturation, and that the inhibition of ROCK activity has profound effects on oocyte meiotic progression and polar body extrusion. Our findings further suggest that short-term inhibition of ROCK for 4 h pre-IVM in a biphasic IVM approach improves oocyte nuclear maturation, producing more MII oocytes, possibly as a result of modulating the activities of several downstream molecules that regulate the expression of cytokinesis and inhibit apoptosis. This positive result suggests ROCK inhibitor as a potential candidate molecule to exploit in the control of oocyte meiotic maturation.

## Figures and Tables

**Figure 1 animals-10-00750-f001:**
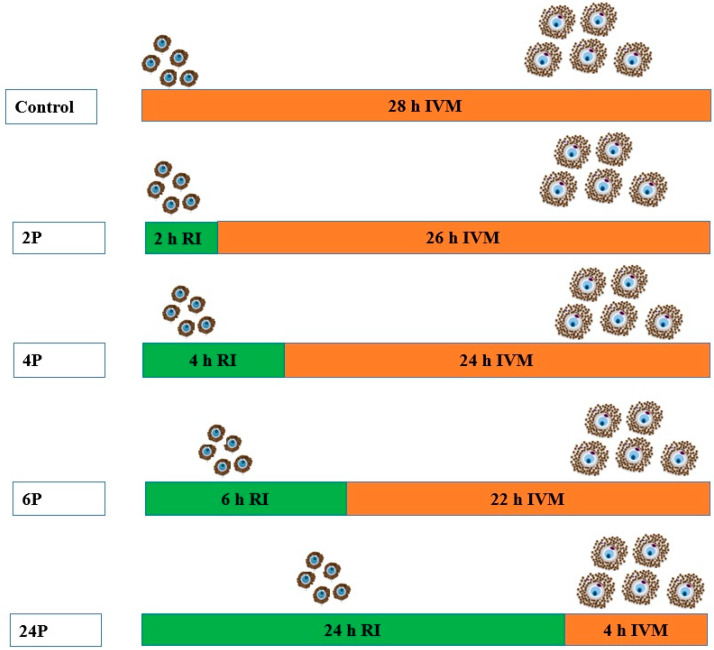
The experimental design of biphasic in vitro maturation (IVM) and prematuration (Pre-IVM). Phase 1: Cumulus–oocyte complex (COC) in vitro maturation medium was supplemented with Rho-associated protein kinases (ROCK) inhibitor (Y-27632) for 2 h, 4 h, 6 h, and 24 h vs. control group without RI. Phase 2: COCs were transferred to plain IVM medium to complete the maturation for 28 h (i.e., 26 h, 24 h, 22 h, and 4 h, respectively).

**Figure 2 animals-10-00750-f002:**
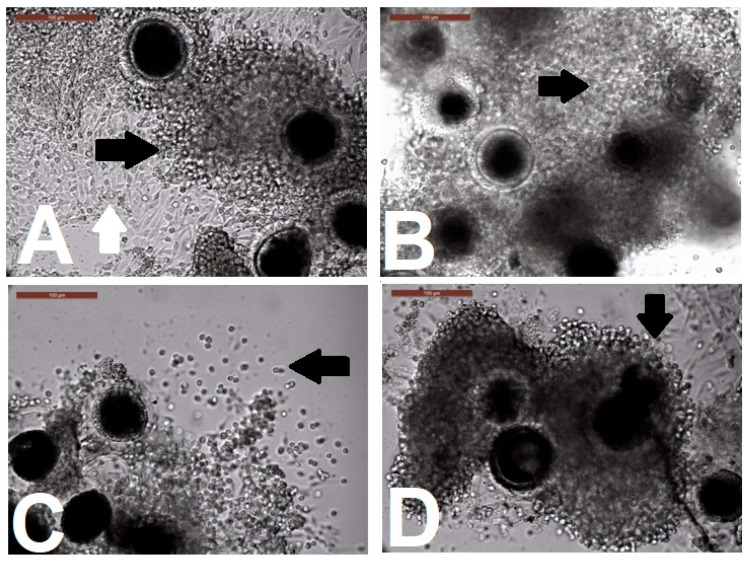
Different degrees of cumulus expansion after biphasic in vitro maturation with ROCK-inhibitor supplementation pre-IVM. (**A**,**B**,**D**) display partial expansion, grade 1 for control, 2RI, and 6RI biphasic IVM, respectively. (**C**) shows full cumulus expansion, grade 2, observed in the 4RI biphasic IVM group. Arrows show the transparency of intercellular spaces between the cumulus cells. The white arrow in A shows the compactness of cumulus cells after biphasic IVM. The arrow in C shows the fully expanded corona radiata, the innermost part of the cumulus cell. Scale bar = 100 µm.

**Figure 3 animals-10-00750-f003:**
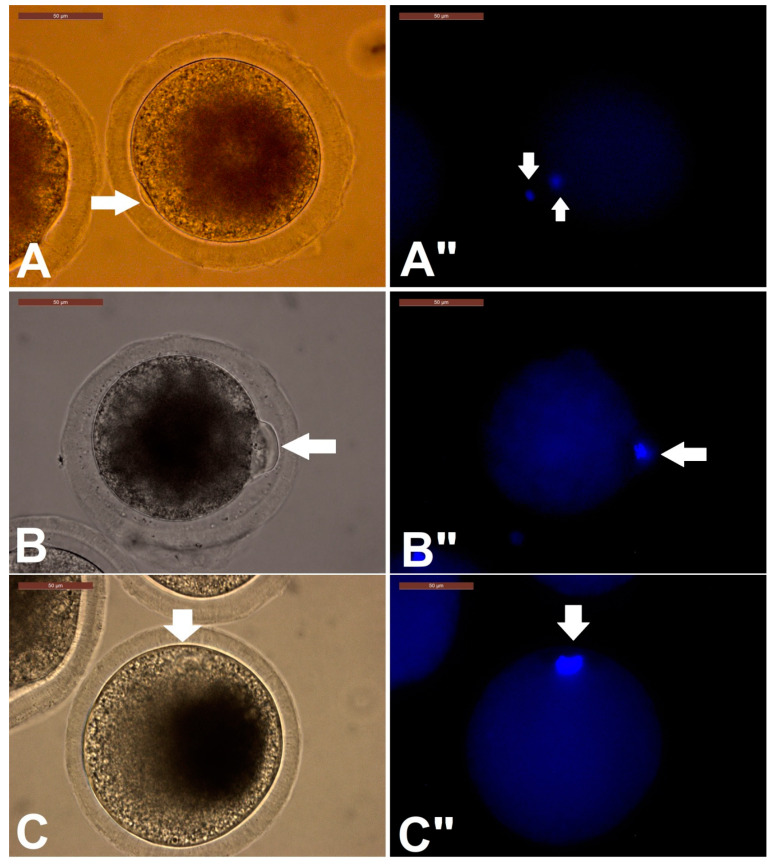
Patterns of nuclear maturation after biphasic IVM. First polar body extrusion after biphasic IVM. Staining was done with Hoechst stain and visualized under fluorescence microscope. (**A**) First polar body extrusion and metaphase II oocyte. The arrow in A indicates a polar body and its location beside the metaphase II plate. (**A’’**) is the corresponding fluorescent image of A. (**B**) Oocyte with large polar body, indicating impaired cytokinesis. (**B’’**) is the corresponding fluorescent image of B. (**C**): Anaphase I. (**C”**) corresponding fluorescent image of C. Scale bar = 50 µm.

**Figure 4 animals-10-00750-f004:**
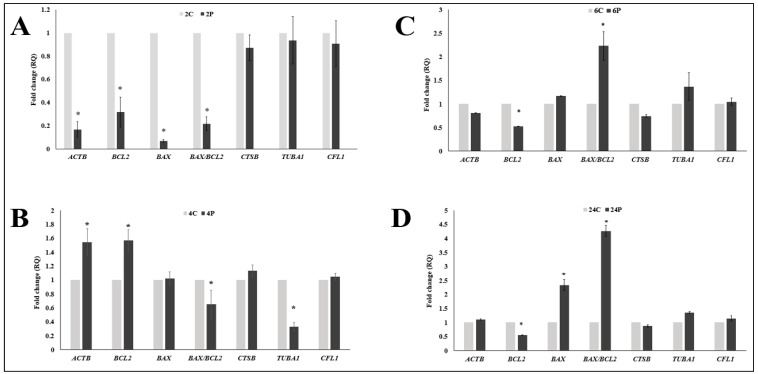
Effect of ROCK-inhibitor supplementation on phase I, pre-IVM for 2 h, 4 h, 6 h, and 24 h on the cytokinesis- and apoptosis-related mRNA transcripts of camel COCs. (**A**–**D**) indicate the treatment groups 2 h, 4 h, 6 h and 24 h respectively. Values were compared to the control groups (without ROCK- inhibitor). Asterisks (*) indicate significant differences (*p* < 0.05) between the values. *ACTB*: beta (β)-actin protein gene, *BCL2*: B-cell lymphoma-2 (Apoptosis regulator, antiapoptotic), *BAX*: Proapoptotic, *BAX*/BCL2: shows the susceptibility to death by apoptosis. The ratio of CTSB: Encodes cathepsin B enzyme which induces apoptosis through the stimulation of Caspase 3, *TUBA1*: Encodes alpha-tubulin (α-tubulin), *CFL1*: Encodes cofilin. Gray bars represent the control group, while the dark ones represent the RI-supplemented group.

**Figure 5 animals-10-00750-f005:**
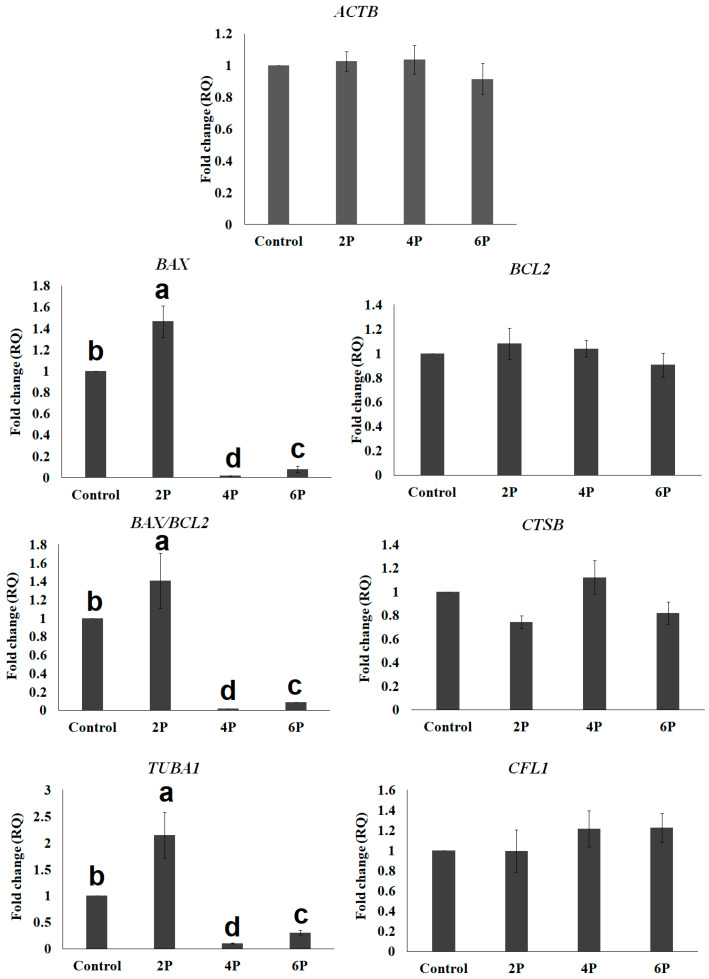
Effect of pre-IVM ROCK-inhibitor supplementation (2 h, 4 h, and 6 h) on the cytokinesis- and apoptosis-related mRNA transcripts of mature camel COCs. Values were compared to the control groups (without ROCK inhibitor). Values carrying different letters (a, b, c, and d) indicate significant differences (*p* < 0.05) between the values. *ACTB*: beta (β)-actin protein gene, *BCL2*: B-cell lymphoma-2 (Apoptosis regulator, antiapoptotic), *BAX*: Proapoptotic, *BAX*/BCL2: shows the susceptibility to death by apoptosis, *CTSB*: Encodes cathepsin B enzyme which induces apoptosis through the stimulation of Caspase 3, *TUBA1*: Encodes alpha-tubulin (α-tubulin), *CFL1*: Encodes cofilin.

**Table 1 animals-10-00750-t001:** Primer information used for real-time PCR.

Gene	Sequence (5′−3′)	Accession No.	Fragment Size (bp)
*BLC2*	F: TGGATCCAGGATAACGGAGG	XM_010979993.1	92
	R: TTCAGAGACAGCCAGGAGAAA		
*BAX*	F:CACCAAGGTGCCTGAACTGA	XM_010996357.1	130
	R: CGTGGGTGTCCCAAAGTAGG		
*CTSB*	F: CAGATGATTGGCAGATGGGC	XM_010995147.1	90
	R: CTTCGCTGATCCTCGGTCTC		
*ACTB*	F: ATCTGGCACCACACCTTCT	XM_010997926.1	137
	R: GGGGTGTTGAAGGTCTCGAA		
*TUBA1A*	F: GGAGACCTGGCCAAAGTACA	XM_010997479.1	95
	R: CAGGCTTTTCCAGTGTGACG		
*CFL1*	F: ACGCCACCTATGAGACCAAG	XM_010995981.1	111
	R: CATCCTTGGAGCTGGCATAG		
*GAPDH*	F: TGCTGAGTACGTTGTGGAGT	XM_010990867	134
	R: TCACGCCCATCACAAACATG		

**Table 2 animals-10-00750-t002:** Effect of ROCK inhibitor on camel cumulus expansion and oocyte in vitro maturation.

Parameter	Control	2P	4P	6P	24P
Number of IVM oocyte	75	75	75	75	75
Degenerated oocyte	36.0 ± 2.2 ^ab^	38.6 ±1.6 ^ab^	17.3 ±2.2 ^b^	34.7 ± 1.2 ^ab^	44 ± 0.8 ^a^
(%)	(27)	(29)	(13)	(26)	(33)
No polar body	37.33 ± 0.9 ^a^	35.66 ± 1.6 ^a^	28.0 ± 2.4 ^a^	40 ± 0.2 ^a^	52 ± 0.5 ^a^
(%)	(28)	(38)	(21)	(30)	(39)
First polar body	26.6 ± 2.8 ^b^	25.57± 1.6 ^b^	54.67± 4.6 ^a^	25.3 ± 2.0 ^b^	4 ± 0 ^b^
(%)	(20)	(8)	(41)	(19)	(3)
Cumulus expansion degree	1 ± 0 ^b^	1 ± 0 ^b^	2 ± 0 ^a^	1 ± 0 ^b^	0 ± 0 ^c^

2P, 4P, 6P, and 24P are the COCs treated with ROCK inhibitor for 2 h, 4 h, 6 h, 24 h, respectively. Data represent means of percentages of five replicates ± standard error of means (SEM). Values between brackets are the total number of the replicates. Values carrying different superscripts within the same row are statistically significant at *p* < 0.05.

**Table 3 animals-10-00750-t003:** Effect of ROCK inhibitor on camel oocyte morphometrics.

	No	Oocyte Diameter (µm)	Zona Pellucida Thickness (µm)	Perivitelline Space Length (µm)	Ooplasm Diameter (µm)
2P	25	156.28 ± 2.75 ^a^	11.40 ± 0.35 ^b^	6.01 ± 0.30 ^a^	116.43 ± 1.9 ^a^
4P	25	151.55 ± 1.76 ^a^	10.37 ± 0.34 ^bc^	3.19 ± 0.30 ^b^	116.67 ± 1.2 ^a^
6P	25	138.82 ± 1.11 ^b^	9.14 ± 0.25 ^c^	6.55 ± 0.7 ^a^	103.93 ± 0.7 ^b^
Control	25	153.09 ± 1.83 ^a^	12.97 ± 0.5 ^a^	2.56 ± 0.41 ^b^	119.23 ± 2.1 ^a^

2P, 4P, and 6P are the COCs treated with ROCK inhibitor for 2, 4, and 6 h, respectively. Values carrying different superscripts within the same raw are statistically significant at *p* < 0.05.

**Table 4 animals-10-00750-t004:** Pearson’s correlation coefficient (r) between oocyte parameters and mRNA transcripts’ expression.

Parameter	Maturation	Degeneration	Cumulus	Oocyte	Ooplasm	ACTB	BAX	BCL2	BAX/BCL2	CFLN Exp.	TUBA Exp.	CTSB Exp.
	%	%	Expansion	Diameter	Diameter	Exp.	Exp.	Exp.				
**Maturation %**	1.00											
**Degeneration %**	−0.83	1.00										
**Cumulus** **Expansion**	0.85	−0.99	1.00									
**Oocyte diameter**	−0.08	0.03	0.13	1.00								
**Ooplasm diameter**	−0.14	−0.10	0.23	0.94	1.00							
**ACTB expression**	0.65	−0.66	0.77	0.68	0.65	1.00						
**BAX expression**	−0.56	0.70	−0.58	0.72	0.55	0.06	1.00					
**BCL2 expression**	0.01	0.03	0.13	0.98	0.87	0.72	0.73	1.00				
**BAX/BCL2**	−0.59	0.71	−0.58	0.72	0.56	0.05	1.00	0.73	1.00			
**CFLN expression**	0.84	−0.59	0.52	−0.59	−0.66	0.13	−0.74	−0.48	−0.77	1.00		
**TUB expression**	−0.47	0.77	−0.65	0.56	0.31	−0.03	0.96	0.61	0.95	−0.55	1.00	
**CTSB expression**	0.05	−0.57	0.58	0.36	0.65	0.45	−0.22	0.23	−0.19	−0.30	−0.49	1.00

r values > 0.7 and > 0.5 indicate strong and moderate correlation, respectively. Exp. means expression.

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
