# Peer review of "Effects of Short-Term Inhibition of Rho Kinase on Dromedary Camel Oocyte In Vitro Maturation"

_animals, 2020, doi:10.3390/ani10050750_

Round 1
Reviewer 1 Report
The present version of article intitled “Effects of short-term inhibition of Rho Kinase on dromedary camel oocyte in vitro maturation” is focused on the efficiency of inhibition for 4 hours pre-UVM using ROCKi to improve the oocyte maturation.
This original work is highlighting various key points about oocyte maturation and the necessity of pre-maturation phase in one hand. In the other hand, it is encouraging the use of ROCKi in oocyte maturation while there is until now lot of hesitation about it since that its effect still unclear.
However, there are some minor revisions that I recommend to improve the quality of this manuscript:
- In introduction:
Line 54-55, page 2: “One major challenge … resumption”. This sentence needs to be rewritten especially that there are many works elucidated about meiotic resumption of camel oocytes.
Line 60-61, page 2: “prematurely … competent”. You are talking here about the lack of synchronization between nuclear maturation and cytoplasmic one.
Line 68, page 2: “blockage of cAMP signals” and “inhibition of key molecules”. Give examples of the used molecules.
Before to talk about ROCK and ROCKi. You need to introduce the applications of ROCKi.
Indeed, ROCKi was recognized to improve the stemness and in the maintenance of embryonic stem cells properties, promoting their recovery and their post-thaw proliferation. Recently, ROCKi could be implemented in in vitro production of embryos specially to increase the revivability of blastocysts after vitrification and even for oocytes. Few studies could to show the efficiency of ROCKi in maturation medium of oocytes in some animal species.
- In Materials and Methods
Line 110, page 3: delete the expression “at least”. why at least? If we try to calculate the number of COCs of each group based on the given design, we will have (15 COC X 5) X 5 = 375. So, we cannot have more assays based on what you gave as number of 375 of total COCs! Indeed, each group contained 75 COCs.
Line 119, page 3: correct “morphometric” by “morphometric evaluation”.
Line 128, page 4: correct “mechanically” by “chemically”. It's not just mechanically since there is use if hyaluronidase. So, it's enzymatic denudation.
Line 143- 165, page 4: The experimental design for the second experience (phase 1 and 2) still unclear and needs to make clear chart of how COC samples were divided in different groups for pre-IVM and post-IVM.
- In Results:
Line 207, page 5: “Table 2”. The results of Cumulus expansion degree are irrelevant. It would be preferable to give the percentage of COCs with degree 1 for each group.
Line 208, page 6: “Figure 2”. Correct “100µ” with “100µm”.
Line 236, page 7: “Figure 3”. Give the Scale bar.
Line 264, page 9: “Figure 4”. Mention that gray color is indicating the control group while the dark one is with RI supplementation. These results are of phase 1 of Experience 2 to show the effect in pre-IVM? If Yes, add this note in the title.
Line 290, page 10: “Figure 5”. This Figure needs to be homogeneous with Figure 4.
- In Discussion:
Line 330, page 13: Correct “ Fertilisation” with “ Fertilization”
Line 347, page 13: Delete “And” in the beginning of the sentence of “ as expected …”
Line 351, page 13: “ oocyte morphometry”. What about zona pellucida thickness which is correlated with premature Ca2+ signal, and oocyte diameter which is correlated with the direct effect of ROCKi on cytoskeleton highlighting different oocyte abnormalities? This issue needs to be developed.
Line 356, page 13: “with reduced oocyte competence”. You need to explain that is due to loss of gap junctions between oocyte and cumulus cells!
Discussion needs to be more developed explaining deeper the role of ROCKi especially in inhibition of apoptosis and to repair the actin cytoskeleton if it is affected. Moreover, ROCKi increased cellular microtubule acetylation since that ROCK regulates microtubule acetylation via phosphorylation of the tubulin polymerization-promoting protein 1.
In the other, how can you explain the results of (An et al., 2018) using ROCKi in maturation medium for 24h of goat oocytes having an decrease in maturation rate from 45% (Control) to 42% but when there is special combination of Cys/LIF with ROCKi, the maturation rate was improved to reach 67%, knowing that the oocyte maturation between species is different?
An, L., Liu, J., Du, Y., Liu, Z., Zhang, F., Liu, Y., ... & Li, Y. (2018). Synergistic effect of cysteamine, leukemia inhibitory factor, and Y27632 on goat oocyte maturation and embryo development in vitro. Theriogenology, 108, 56-62.
Another study of Arayatham et al. (2017) reported that ROCKi at 10µM in maturation medium did not compromise the meiotic developmental competence of feline oocytes, instead it improved the cytoplasmic maturation post-thawed oocytes. How can you explain that comparing it with your results?
Arayatham, S., Tiptanavattana, N., & Tharasanit, T. (2017). Effects of vitrification and a Rho-associated coiled-coil containing protein kinase 1 inhibitor on the meiotic and developmental competence of feline oocytes. Journal of Reproduction and Development.
Moreover, what is the difference between the use of ROCKi and the different used molecules in pre-IVM in term of maturation rate?
Author Response
The present version of article intitled “Effects of short-term inhibition of Rho Kinase on dromedary camel oocyte in vitro maturation” is focused on the efficiency of inhibition for 4 hours pre-UVM using ROCKi to improve the oocyte maturation.
This original work is highlighting various key points about oocyte maturation and the necessity of pre-maturation phase in one hand. In the other hand, it is encouraging the use of ROCKi in oocyte maturation while there is until now lot of hesitation about it since that its effect still unclear.
Response: We appreciate the time, efforts, and overall constructive suggestions and comments raised by the reviewer that greatly contributed to improving the manuscript quality. All the suggestions have been addressed, editing and corrections were made accordingly.
However, there are some minor revisions that I recommend to improve the quality of this manuscript:
- In introduction:
Line 54-55, page 2: “One major challenge … resumption”. This sentence needs to be rewritten especially that there are many works elucidated about meiotic resumption of camel oocytes.
R: Thank you for your suggestion. We modified the text accordingly.
Line 60-61, page 2: “prematurely … competent”. You are talking here about the lack of synchronization between nuclear maturation and cytoplasmic one.
R: Yes, we agree with the reviewer. We modified the text for better understanding.
Line 68, page 2: “blockage of cAMP signals” and “inhibition of key molecules”. Give examples of the used molecules.
R: Thanks for the suggestion. We added examples accordingly.
Before to talk about ROCK and ROCKi. You need to introduce the applications of ROCKi.
Indeed, ROCKi was recognized to improve the stemness and in the maintenance of embryonic stem cells properties, promoting their recovery and their post-thaw proliferation. Recently, ROCKi could be implemented in in vitro production of embryos specially to increase the revivability of blastocysts after vitrification and even for oocytes. Few studies could to show the efficiency of ROCKi in maturation medium of oocytes in some animal species.
R: We thank you for the suggestion. We add this part accordingly.
- In Materials and Methods
Line 110, page 3: delete the expression “at least”. why at least? If we try to calculate the number of COCs of each group based on the given design, we will have (15 COC X 5) X 5 = 375. So, we cannot have more assays based on what you gave as number of 375 of total COCs! Indeed, each group contained 75 COCs.
R: Thank you for this insightful note. The text has been modified.
Line 119, page 3: correct “morphometric” by “morphometric evaluation”.
R: We corrected the text.
Line 128, page 4: correct “mechanically” by “chemically”. It's not just mechanically since there is use if hyaluronidase. So, it's enzymatic denudation.
R: Thank you for this insightful note, we corrected the text accordingly.
Line 143- 165, page 4: The experimental design for the second experience (phase 1 and 2) still unclear and needs to make clear chart of how COC samples were divided in different groups for pre-IVM and post-IVM.
R: We have edited the figure legend and the text for better understanding.
- In Results:
Line 207, page 5: “Table 2”. The results of Cumulus expansion degree are irrelevant. It would be preferable to give the percentage of COCs with degree 1 for each group.
R: We thank the reviewer for this notion. We found a homogenous expansion for each group and as you know the cumulus cells from different COCs intermingled to each other to make a clump-like collection. We considered the whole patch of COCs for the measurement of cumulus expansion degree. We hope our response clarifies the inquiry.
Line 208, page 6: “Figure 2”. Correct “100µ” with “100µm”.
R: Corrected.
Line 236, page 7: “Figure 3”. Give the Scale bar.
R: Scale bar was provided.
Line 264, page 9: “Figure 4”. Mention that gray color is indicating the control group while the dark one is with RI supplementation. These results are of phase 1 of Experience 2 to show the effect in pre-IVM? If Yes, add this note in the title.
R: Thank you for this suggestion. We modified the text accordingly.
Line 290, page 10: “Figure 5”. This Figure needs to be homogeneous with Figure 4.
R: We thank the reviewer for this notion. We edited the font, italics, and the size of the figure accordingly.
- In Discussion:
Line 330, page 13: Correct “ Fertilisation” with “ Fertilization”
R: Corrected.
Line 347, page 13: Delete “And” in the beginning of the sentence of “ as expected …”
R: Corrected.
Line 351, page 13: “ oocyte morphometry”. What about zona pellucida thickness which is correlated with premature Ca2+ signal, and oocyte diameter which is correlated with the direct effect of ROCKi on cytoskeleton highlighting different oocyte abnormalities? This issue needs to be developed.
R: Thank you for the useful suggestion. We added some discussion about these interesting effects.
Line 356, page 13: “with reduced oocyte competence”. You need to explain that is due to loss of gap junctions between oocyte and cumulus cells!
R: We thank the reviewer for the insightful suggestions. The text has been modified accordingly.
Discussion needs to be more developed explaining deeper the role of ROCKi especially in inhibition of apoptosis and to repair the actin cytoskeleton if it is affected. Moreover, ROCKi increased cellular microtubule acetylation since that ROCK regulates microtubule acetylation via phosphorylation of the tubulin polymerization-promoting protein 1.
R: We thank the reviewer for this great suggestion. We added some paragraphs regarding the effects on tubulin and apoptosis in the discussion accordingly.
In the other, how can you explain the results of (An et al., 2018) using ROCKi in maturation medium for 24h of goat oocytes having an decrease in maturation rate from 45% (Control) to 42% but when there is special combination of Cys/LIF with ROCKi, the maturation rate was improved to reach 67%, knowing that the oocyte maturation between species is different?
An, L., Liu, J., Du, Y., Liu, Z., Zhang, F., Liu, Y., ... & Li, Y. (2018). Synergistic effect of cysteamine, leukemia inhibitory factor, and Y27632 on goat oocyte maturation and embryo development in vitro. Theriogenology, 108, 56-62.
R: We thank the reviewer for this discussion and suggestion. We added some discussion about this finding in the revised text.
Another study of Arayatham et al. (2017) reported that ROCKi at 10µM in maturation medium did not compromise the meiotic developmental competence of feline oocytes, instead it improved the cytoplasmic maturation post-thawed oocytes. How can you explain that comparing it with your results?
Arayatham, S., Tiptanavattana, N., & Tharasanit, T. (2017). Effects of vitrification and a Rho-associated coiled-coil containing protein kinase 1 inhibitor on the meiotic and developmental competence of feline oocytes. Journal of Reproduction and Development.
R: We thank the reviewer for this insightful discussion. However, when we read the paper of Arayatham et al., we found that ROCKi caused increased MI oocytes (29% vs 20% in control, P < 0.05), while it decreased MII oocytes (46% vs 53 % in control, P > 0.05), and High concentrations of the inhibitor (20 µM and 40 µM) significantly decreased meiotic competence, which appeared to be caused by a high incidence of MI arrest. Their results support our finding, even the effect on MII oocytes was not significant in their results, but the improvement in cleavage might be due to the stabilizing effects of RI on the cytoplasm of vitrified oocytes, but not on non-vitrified ones. We highlighted this in the discussion.
Moreover, what is the difference between the use of ROCKi and the different used molecules in pre-IVM in term of maturation rate?
R: Actually this effect will vary among the species and there are great efforts and experiments performed by Robert Gilchrist and his group and other groups on these molecules in humans and bovine. For example, in humans, MII was increased to 62% vs. 47.9 % when using c-type natriuretic peptide (Sanchez, F.; Le, A.H.; Ho, V.N.A.; Romero, S.; Van Ranst, H.; De Vos, M.; Gilchrist, R.B.; Ho, T.M.; Vuong, L.N.; Smitz, J. Biphasic in vitro maturation (CAPA-IVM) specifically improves the developmental capacity of oocytes from small antral follicles. Journal of Assisted Reproduction and Genetics 2019, 36, 2135-2144, doi:10.1007/s10815-019-01551-5). A similar trend of the increase was found in the studies that we have cited (Ref 16 to 25).
----
Thank you for your comments and suggestions that greatly appreciated.
Reviewer 2 Report
In the present study the authors evaluated the effects of ROCK inhibitor (Y-27632) in a biphasic IVM of dromedary camel oocytes on cumulus cell expansion, nuclear maturation, oocyte morphometric and mRNA transcript of selected apoptotic (BCL2, 149 BAX, and CTSB) and cytokinesis related genes (ACTB, TUBA1A, and CFL1). The authors concluded that ROCK is involved in dromedary camel oocyte maturation, and the inhibition of ROCK activities had profound effects on oocyte meiotic progression and polar body extrusion. The study is well designed and provides new approaches to improve the efficiency of oocyte maturation in dromedary camel. However, before the manuscript accepted for publication few changes should be considered
- Line 23, remove is after was
- Line 110, remove at least. According to the experimental design and the number of oocytes used in each group (Each group contained 15 COCs) and the total number of oocytes (375) used in this experiment, so I think 5 replicates is the maximum
- Table 2, please, clarify what are the numbers between brackets with ± SEM representing for. Also, indicate this explanation under the table and indicate also that the numbers after ± stand for SEM.
- Under the table, line 209, Value should be Values and write that (Values carrying different superscripts within the same raw are statistically significant at P < 0.05). Same changes should be done under table 3
- Line 394 (ROCK is in involved) remove in before involved
- Lines 381-388, please highlight in details the role of ROCK signaling in BAX/BCL2 gene expression or what is the importance of evaluating the apoptosis related genes in the current study
- It will a good addition for this paper if the authors are able to present some measurements for spindle morphology and β-actin status in oocytes after IVM through immunocytochemistry
- Please, could you clarify in the methodology section or in the result section if the oocytes that used for evaluation of oocyte morphometric (25 oocytes in each group as indicated in table 3 are included in the total numbers of oocyte used in this experiments
Reference:
Reference 10 is wrong (Sun, Q.-Y.; Fathi, M.; Moawad, A.R.; Badr, M.R. Production of blastocysts following in vitro maturation 443 and fertilization of dromedary camel oocytes vitrified at the germinal vesicle stage. PloS one 2018, 13, 444 e0194602, doi:10.1371/journal.pone.0194602). Sun, Q-Y is not an author in this paper Reference 8 and 32 are the same Reference 6: No journal includedAuthor Response
Response: We acknowledge the time, efforts, and comments by the reviewer that greatly contributed to improving the manuscript quality. We substantially revised the manuscript according to your suggestion.
- Line 23, remove is after was
R: We are sorry for this error. We deleted it.
- Line 110, remove at least. According to the experimental design and the number of oocytes used in each group (Each group contained 15 COCs) and the total number of oocytes (375) used in this experiment, so I think 5 replicates is the maximum
R: Thank you for your insightful note. We corrected the text.
- Table 2, please, clarify what are the numbers between brackets with ± SEM representing for. Also, indicate this explanation under the table and indicate also that the numbers after ± stand for SEM.
R: Thank you for this suggestion. We corrected the text accordingly.
- Under the table, line 209, Value should be Values and write that (Values carrying different superscripts within the same raw are statistically significant at P < 0.05). Same changes should be done under table 3
R: Thank you for this correction. We modified the text accordingly.
- Line 394 (ROCK is in involved) remove in before involved
R: We apologize for the mistyping. It has been corrected.
- Lines 381-388, please highlight in details the role of ROCK signaling in BAX/BCL2 gene expression or what is the importance of evaluating the apoptosis related genes in the current study
R: Thank you for this suggestion. We added some text as recommended.
- It will a good addition for this paper if the authors are able to present some measurements for spindle morphology and β-actin status in oocytes after IVM through immunocytochemistry
R: We thank you for this very useful suggestion. However, we faced some problems with antibodies valid for camel proteins even we tried for some proteins such as for Vimentin in our recent report (Saadeldin et al. 2018 Cumulus cells of camel (Camelus dromedarius) antral follicles are multipotent stem cells. Theriogenology 118: 233-242). The trials for using the commercially available antibodies for immunofluorescence or Western blot for detecting beta-actin was unsuccessful and we are in need for trying several commercially available antibodies to detect their cross-reactivity with camel proteins. We had the trial to stain meiotic spindles in porcine too (Saadeldin et al. 2016. Blastocysts derivation from somatic cell fusion with premature oocytes (prematuration somatic cell fusion).Dev Growth Differ 58(2):157-66), but we need to apply the specific antibodies for camel. We hope you consider this shortage that is still out of our control.
Please, could you clarify in the methodology section or in the result section if the oocytes that used for evaluation of oocyte morphometric (25 oocytes in each group as indicated in table 3 are included in the total numbers of oocyte used in this experiments
R: Thank you for this inquiry. The morphometric evaluation was performed at the same oocytes were used for meiotic evaluation but with selected oocytes (n=25). Images were captured and analyzed first for nuclear status and then for the morphometric parameters.
Reference:
Reference 10 is wrong (Sun, Q.-Y.; Fathi, M.; Moawad, A.R.; Badr, M.R. Production of blastocysts following in vitro maturation 443 and fertilization of dromedary camel oocytes vitrified at the germinal vesicle stage. PloS one 2018, 13, 444 e0194602, doi:10.1371/journal.pone.0194602). Sun, Q-Y is not an author in this paper Reference 8 and 32 are the same Reference 6: No journal included
R: We are sorry for the defect of Endnote software. We usually face this problem with the papers of Plos One journal, Endnote software writes the Editor’s name as the first author. We revised all references accordingly and corrected the missed parts.
-----
We appreciate the reviewer for the great comments.
Reviewer 3 Report
The research add new information on dromedary camel oocyte in vitro maturation. The aim is clear, methods appropriate, results are discussed properly. The manuscript is detailed, but English language should be revised.
Some suggestions (only some examples are reported, all the text should be checked):
- Line 49 WERE instead of "was"
- Line 50 "reviewd by [...]" delete by or write the Authors of the references
- Line 51: "even at that" is not used in the right way. Rephrase
- Line 53: ARE instead of "is"
- Line 75: delete "Since" or connect the sentence to the subsequent one
- Lines 243-245: ZONA pellucida instead of "zonal pellucida"
- Line 300 they ARE both correlated
- Line 325-327: "All these studies [ADD REFERENCES] (have)DELETE used varying doseS of meiotic inhibitor (at) FOR different pre-IVM durationS. The procedure used in the present study examined the pre-IVM duration effect using THE same dose of meiotic inhibitor, RI.
- Line 335-345-347: PROLONGED instead of prolong
- Line 343-344: "Since cumulus expansion is positively linked to oocyte maturation [45]. The partial ..." connect the sentences with a comma, otherwise do not use Since in that way.
- Line 349: THESE OBSERVATIONS insted of this observation
- Line 355: "This also corroborates the reports of [46] in...." add Author et al. after of
- Line 376: "Besides,..." it is not used correctly. Rephrase
- Line 380: STAGES instead of stage
Author Response
Response: We appreciate the time, efforts, and suggestions by the reviewer. All the suggestions have been addressed, and corrections made accordingly. In addition, we edited the manuscript and revised the English writing as recommended.
Some suggestions (only some examples are reported, all the text should be checked):
- Line 49 WERE instead of "was"
R: Corrected.
- Line 50 "reviewd by [...]" delete by or write the Authors of the references
R: Corrected accordingly.
- Line 51: "even at that" is not used in the right way. Rephrase
R: Corrected accordingly.
- Line 53: ARE instead of "is"
R: Corrected.
- Line 75: delete "Since" or connect the sentence to the subsequent one
R: Corrected.
- Lines 243-245: ZONA pellucida instead of "zonal pellucida"
R: We are sorry for this mistyping. It has been corrected throughout the manuscript.
- Line 300 they ARE both correlated
R: Corrected.
- Line 325-327: "All these studies [ADD REFERENCES] (have)DELETE used varying doseS of meiotic inhibitor (at) FOR different pre-IVM durationS. The procedure used in the present study examined the pre-IVM duration effect using THE same dose of meiotic inhibitor, RI.
R: We thank you for these corrections. We corrected the text accordingly.
- Line 335-345-347: PROLONGED instead of prolong
R: Corrected.
- Line 343-344: "Since cumulus expansion is positively linked to oocyte maturation [45]. The partial ..." connect the sentences with a comma, otherwise do not use Since in that way.
R: It has been corrected.
- Line 349: THESE OBSERVATIONS insted of this observation
R: Corrected.
- Line 355: "This also corroborates the reports of [46] in...." add Author et al. after of
R: Corrected.
- Line 376: "Besides,..." it is not used correctly. Rephrase
R: “Meanwhile” was added.
- Line 380: STAGES instead of stage
R: Corrected.